# Design and Manufacture of an Optimised Side-Shifted PPM EMAT Array for Use in Mobile Robotic Localisation

**DOI:** 10.3390/s23042012

**Published:** 2023-02-10

**Authors:** Ross McMillan, Rory Hampson, Morteza Tabatabaeipour, William Jackson, Dayi Zhang, Konstantinos Tzaferis, Gordon Dobie

**Affiliations:** Centre of Ultrasonic Engineering, University of Strathclyde, Glasgow G1 1XQ, UK

**Keywords:** PPM EMATs, guided waves, ranging and mapping, shear horizontal, unidirectional

## Abstract

Guided wave Electro Magnetic Acoustic Transducers (EMATs) offer an elegant method for structural inspection and localisation relative to geometric features, such as welds. This paper presents a Lorentz force EMAT construction framework, where a numerical model has been developed for optimising Printed Circuit Board (PCB) coil parameters as well as a methodology for optimising magnet array parameters to a user’s needs. This framework was validated experimentally to show its effectiveness through comparison to an industry built EMAT. The framework was then used to design and manufacture a Side-Shifted Unidirectional Periodic Permanent Magnet (PPM) EMAT for use on a mobile robotic system, which uses guided waves for ranging to build internal maps of a given subject, identifying welded sections, defects and other structural elements. The unidirectional transducer setup was shown to operate in simulation and was then manufactured to compare to the bidirectional transmitter and two-receiver configurations on a localisation system. The unidirectional setup was shown to have clear benefits over the bidirectional setup for mapping an unknown environment using guided waves as there were no dead spots of mapping where signal direction could not be interpreted. Additionally, overall package size was significantly reduced, which in turn allows more measurements to be taken within confined spaces and increases robotic crawler mobility.

## 1. Introduction

Robotically driven non-destructive testing (NDT) can be used to safely quantify the health of large structural assets such as storage tankers and vessels. These structures are subject to natural corrosion, decay and cracking, and as such are required to be assessed and maintained in order to continue operating safely [1,2,3,4]. 

A mobile robotic inspection system that is currently under development uses Shear Horizontal (SH) guided waves as the sensory input for mapping structures, as SH guided waves have been shown to reflect back from features such as boundaries, welds and defects [5,6]. In addition, guided waves can travel large distances, compared to bulk ultrasound, from a single point of generation, making them suitable for ranging in mapping and robot localisation [7]. This means that guided waves can be used to inspect inaccessible areas in a structure where direct transducer contact to the area of inspection may not be possible [5,8]. SH waves can be produced using Electromagnetic Acoustic Transducers (EMATs).

Different guided wave types (modes) can be formed in plates. Shear Horizontal guided waves oscillate perpendicular to the direction of propagation in the transverse plane [9] and the SH1 mode was found to reflect highly off commonly found butt weld heads [6], and as such is the mode of focus in this work. Although structures, such as tankers and vessels, are essentially large cylinders, the guided waves generated in this work are calculated for plates, as if the ratio of the radius to thickness is greater than 10:1 the effect of plate curvature on the dispersion curves is negligible [10,11].

EMATs can be simply described as a pulsed coil behind a magnet or an array of magnets; utilizing them is a robust and reliable way to generate guided waves. EMATs are also the only feasible way to generate Shear Horizontal guided waves. As electromagnetism is the mechanism used to generate waves within a subject, EMATs are non-contact, and in the context of robotic crawlers, couplant is not required so umbilical cords are not needed to supply fluid. However, EMATs typically require high voltage drivers to get signals of reasonable amplitude and can have poor SNR (Signal to Noise Ratio, the ratio of signal amplitude and instrumentation noise) compared to other methods of ultrasonic transduction.

In the literature, there is little work on the optimisation and selection of EMAT PCB racetrack coil parameters to improve the issues that EMATs have in producing high amplitude waves with low excitation voltages due to low transduction efficiency. This issue in EMAT design was addressed in this work and was applied to the development of the authors robotic mapping system [6].

### 1.1. PPM EMATs

Permanent Periodic Magnet (PPM) Lorentz Force EMATs are the most practical method of generating SH guided waves, as magnetostrictive methods require bonding and coupling [9,12,13]. Lorentz forces are the main mechanism of generation within a given sample using PPM EMATs [13], detailed in Figure 1. The interaction between the eddy current density, $J_{e}$, induced into the subject by the alternating current in the racetrack coil, and the total magnetic flux density, B, produces Lorentz forces. The first magnetic flux density is the background static magnetic field vector, $\bar{B}$, from the permanent magnet array and the second magnetic flux density is from the dynamic field vector, $\tilde{B}$, generated by the AC current in the racetrack coil. Equations (1)–(3) [13,14,15] give the equations for the Lorentz Force, $F_{L},$ and its components, where $n_{e}$ is electron density, e is the electron charge and $v_{e}$ is the mean electron velocity.
(1)$$J_{e}=-n_{e}ev_{e}$$
(2)$$B=\bar{B}+\tilde{B}$$
(3)$$F_{L}=J_{e}\times B$$

In ferromagnetic materials, magnetisation and magnetostriction occur, as well as Lorentz forces, when using a PPM EMAT. In steel samples, such as those used in this work, magnetostriction is negligible, meaning Lorentz forces followed by magnetisation are the main driving mechanisms for guided wave generation [16]. The magnetisation force is described by Equation (4) [14], where ∇H is the 3 × 3 s-order tensor, which gives the gradient of the magnetic field, $\mu_{0}$ is the magnetic permeability of free space and M is the magnetisation.
(4)$$F_{M}=\nabla H\cdot \mu_{0}M$$

This makes the total dynamic forces applied to a steel body as follows (Equation (5)).
(5)$$F=F_{L}+F_{M}$$

However, it has been shown that magnetisation due to cancelling is negligible [13,17]; therefore, this work is only concerned with the Lorentz force mechanism.

Currently, the guided wave mapping system in [5] makes use of a traditional commercial PPM EMAT transmitter and a receiver pitch-catch setup, where, in order to produce waves with increased amplitudes, high voltages have to be applied to the transmitter and the distance between subject surface and the transducers has to be minimised without making contact with the surface. To generate guided waves with high voltages, a special pulser is required. In the context of a mobile robotic system, this pulser brings the same difficulties of an umbilical cord used for providing coupling fluid for bulk ultrasonic measurements. 

In addition, in [5], the transducer setup is bigger than the robot that drives it and uses two receivers and a bidirectional transmitter. This transmitter simultaneously produces both a forward and backward SH1 wave during generation [6]. In a finite subject, due to bidirectional generation, it is difficult to interpret data in real-time as the propagation path of a received reflection is unknown, so the position of the reflective feature relative to the transducer is ambiguous. A fix for this is to use a two-receiver setup to determine the generation direction of a reflected wave and, subsequently, its travelled path using time of flight methods; however, this greatly increases the size of the transducer setup as well as the magnetic hold force, which can impede robotic crawler movement. Reducing the EMAT size for use on robotic crawlers has been explored in [18]; this was done by reducing the magnet array and coil size of standard PPM EMATs. However, this causes the generated signals to be weak and difficult to interpret at times. Furthermore, there is also susceptibility to dead zones of ranging where signals can return to the receivers at the same time, so the direction of the signals cannot be determined.

A solution to the problems presented would be to utilize a unidirectional PPM EMAT setup, which is optimised for producing guided waves in only one direction. This removes the ambiguity around the direction of the received signal as the initial direction of the wave is always known, so ranging can always take place with no dead zones. Thus, removing the need for the second receiver significantly reduces the size of the transducer setup.

### 1.2. Unidirectional EMATs

Unidirectional EMATs generally operate by altering the traditional setup of a given EMAT style to dampen one side of generation, whereby two sources are excited and offset, causing interference between the generated waves. This results in elimination or extreme dampening of one of the forward or backwards generated waves and constructive interference of the other [19,20]. For SH PPM EMATs, due to the magnet array determining the induced force directions, achieving wave interference presents more challenges than the methods cited.

Kubrusly et al. [21] demonstrate a side-shifted magnet array, which equates to two PPM EMATs merged and offset; the two coils in this EMAT are pulsed 90° apart, resulting in single sided generation with small backward side-lobes. [22] This demonstrates a dual-linear coil array rather than the traditional racetrack design, whereby multiple magnet arrays are individually wrapped in wire from one of two coils, which are pulsed with 90° separation creating single sided generation. The resultant wavelength is, however, doubled, making it more suited to low frequencies [23]. The revised side-shifted design of [21] improved the radial pattern by altering the racetrack coil shape to a loop under increased magnet arrays. 

Thon et al. [24] present work on optimising the magnet array of a PPM EMAT for generating plate SH guided plate waves in curved surfaces, such as a pipe. Different magnet array configurations were investigated where the array that produced the best signals was constructed using magnets that were altered to conform to the curvature of the pipe, reducing the distance between the background magnetic field and the sample. Coil optimisation was not investigated in this work. 

S. L. Huang et al. [25] investigated a unique point focussed unidirectional EMAT, which uses a semi-circle shaped magnet array with side-shifted magnets to achieve unidirectional focussing. Multiple studies have been produced on the optimisation of this focussed PPM EMAT style [26,27,28].

### 1.3. This Work

This work presents an optimisation framework for increasing the Lorentz force output in PPM EMATs. This is done through coil and magnet array optimisation and presents the parameters used to achieve the highest Lorentz force from a given set of EMAT design parameters. This framework will also give the reader some direction on designing their own PPM EMAT. 

Following on, this study will look to apply the optimisation method to a unidirectional side-shifted PPM EMAT, where two coils are pulsed independently with one pulse 90° out of phase from the other, similar to that in [21], except that this study is optimising PCB racetrack coils and magnet arrays with a focus for use in mobile robotics. In addition, a receiver EMAT will be constructed; this will then be packaged together with the transmitter as one small unit for ease of deployment onto a mobile robotic system. Following design and experimental validation of the mobile robot focussed transmitter/receiver setup, its ranging capabilities were evaluated by comparing it to the two-receiver bidirectional transmitter setup. Evaluation was carried out through comparison of the two different setups through their ability to range at all measurement points, and the number of measurement points that can be taken by a given setup. Considerations have been made to reduce the obstruction of robot mobility through reducing the high magnetic strength or large package size.

## 2. Optimisation and Methodology 

In this work, EMAT performance optimisation was achieved by increasing the Lorentz force induced into the subject as much as possible without the use of extremely high voltages or high force permanent magnet arrays, which would be unusable in the context of a small mobile robotic application. The Lorentz force was focused on as it is typically the main generation mechanism when applied to materials such as steel and aluminium [13,29].

To do this, the racetrack coil has been optimised for the maximum induced eddy current, increasing $J_{e}$ and $\tilde{B}$, and the magnet array has been optimised for the appropriate magnetic field vector value, $\bar{B}$.

As SH1 is to be generated in a 10 mm steel plate, the dimensions of the magnet arrays are set to fit a 25 mm wavelength, which generates SH1 at 205 kHz, and the coils were designed to appropriately cover the surface area of the magnet array that sits on top of the coil. 

During the design process, experimental manufacture and use were kept in mind; this was applied to multiple parameters of the design process, as it was preferred to have a design that was transferrable from simulation into practical experimentation without significant design changes.

### 2.1. Transmitter Coil Optimisation

This section describes the numerical model created for optimising the racetrack coil. This method operates by taking a number of coil design parameters and then iteratively working through the possible options for coil trace widths and the number of turns, in order to find the set of values that give the highest induction of eddy currents, $J_{e}$, into the sample, which is done through maximising the strength of the magnetising fields generated by the alternating current in the coil [13,30]. Practically, this means maximising the induced eddy currents, which is achieved through optimising the trace width and trace spacing, maximising power transfer and impedance matching the pulser generator pulsing the coil [31].

The coil optimisation shown here is intended for use on Flexible Printed Circuit Board (FPCB) coils, not hand wound coils. There are multiple benefits to using FPCB coils over hand wound coils, which include the following: high coil flexibility easing construction; greater influence over wire/trace parameters to control performance; increased wire/trace to sample surface contact, which increases eddy current transduction; precise manufacturing; and a low profile [13].

The input parameters for the numerical model are as follows:

Coil Dimensions, shown in Figure 2:

$C_{length}$—Total coil end to end length (mm);

$C_{width}$—Total coil width (mm); and 

$M_{width}$—Width of magnet that is to be covered (mm).

PCB Manufacturing Inputs, typically set by manufacturer:

$G_{minimum}$—Minimum gap (mm) between traces;

$T_{minimum}$—Minimum manufacturers trace width (mm);

$W_{Cu}$—Copper Weight (oz/ft^2^);

$P_{thickness}$—PCB Thickness (mm); and

$G_{sample}$—Distance from sample (mm).

Experiment Variables, parameters regarding pulsing:

$V_{in}$—Voltage at pulser source (V);

$Z_{source}$—Impedance at pulser source (Ω);

f—Intended frequency of pulsing (kHz); and

$S_{M}$—Sample Material (requires inductance vs. distance plot).

The coil dimensional parameter inputs were designed to match the EAGLE (Autodesk, Inc. CA, USA) User Language Programme (ULP) that was used to generate the files for manufacturing the PCB coils; this ULP is a spiral coil generator [32]. This eased transfer of coil dimensions output by the model into the manufacturing process. The model operates in the following way.

Following the input of the parameters listed, the applicability of the skin effect is checked for using Equation (6), where $\rho_{Cu}$ and $\mu_{Cu}$ are, respectively, the resistivity and permeability of copper, and $\mu_{0}$ is the permeability constant. If the frequency of the signal causes the skin equation to take effect, the model will stop running. However, the operating frequencies of EMATs are well below the values where the skin effect, $S_{E},$ applies.
(6)$$S_{E}=\sqrt{\frac{\rho_{Cu}}{\pi f\mu_{0}\mu_{Cu}}}$$

The model then iterates, through a number of possible turns, N, within the set coil dimensions until the trace width, $T<T_{minimum}$. For each iteration of N, the following steps are taken. 

The value for $G_{minimum}$ for each iteration is set to the largest value from either the IPC2221 lookup table [33], based on the input voltage, or the minimum manufacturers gap. The trace width, T, required to meet the value of $M_{width}$ for an iteration is calculated using Equation (7).
(7)$$T=\frac{\left(M_{width}-G_{minimum}\left(N-1\right)\right)}{N}$$

Following the process in [34], the critical dimensions are then calculated for the coil based on $C_{length}$, $C_{width}$ and $M_{width}$. This process allows the coil to be generated as a series of end connected line segments. In the case of this study, the Track Width Ratio (TWR) factor is set to 1 to maintain a constant trace width through each turn. The lengths of each line segment in each turn of the coil can be calculated and parallel segments are grouped together for analysis. The total trace length, $T_{TL},$ is the sum of all trace segment lengths, where $T_{AL}$ is the average turn length, $T_{TL}/N$.

Inductance is then calculated for each iteration of N in the following way. Ref [35] gives a method for calculating $L_{SI}$, the summation of the self-inductance of the coil segments. $L_{MI}$ is double the summation of mutual inductances of each line segment to all other line segments calculated following the method in Section 3 of [34]; doubled summation is utilised as symmetry is used in the calculation. The inductance of the coil, L, is calculated using Equation (8). The inductive reactance, $X_{L}$, is calculated using 2πfL.
(8)$$L=L_{SI}+L_{MI}$$

The capacitance of each iteration of N is measured between the input and ground return, given by Equation (9) from [36], using the case of air and FR4 substrate, a common substrate used in PCB manufacture, where $\varepsilon_{Air}$ is the permittivity for air, $\varepsilon_{FR4}$ is the permittivity for FR4 substrate and $\varepsilon_{0}$ is the permittivity of the vacuum. The capacitive reactance, $X_{C}$, is then calculated using 1/2πfC.
(9)$$C=N\frac{T^{2}\varepsilon_{0}\varepsilon_{FR4}}{P_{thickness}}+\frac{\left(0.9\varepsilon_{Air}+0.1\varepsilon_{FR4}\right)\varepsilon_{0}W_{Cu}T_{TL}}{NG_{minimum}}$$

The resistance of each iteration of N is then calculated using Equation (10) from the total length of the coil, $C_{TL}$.
(10)$$R=\frac{\rho_{Cu}T_{TL}}{W_{Cu}T}$$

The impedance of each iteration of the coil is calculated using a planar coil model [36] shown by Equation (11). This impedance value is then adjusted to account for the impedance of air, the impedance of the material under inspection and the EMATs distance from its surface. As the relationship between the coil impedance in free space and the impedance in proximity to a metal sample is non-trivial, an empirical correction factor, *α*, is applied to approximate this relationship. The variable *α* is modelled as a function of the standoff distance, *G_sample_*, and is unique to the sample used in this experiment due to sensitivity to sample properties, such as thickness and permeability.
(11)$$Z=\alpha \left(G_{Sample}\right)\frac{\left(R+X_{L}\right)X_{C}}{R+X_{L}+X_{C}}$$

I=V/Z is then used to estimate the current. Using the current, the magnetisation, $M_{N}$, of each iteration of N can be calculated with $M_{N}=NI$. Thus, the optimum number of turns and trace width can be established through finding which iteration of N produces the highest magnetisation from the values of the input parameters; the highest magnetisation will come from $Z=Z_{Source}$. The model outputs a curve of magnetisation vs. number of turns as well as the optimal trace width and number of turns that are used as ULP inputs to generate the FPCP coil.

### 2.2. Transmitter Magnet Array Optimisation

To optimise the magnet array of a PPM EMAT being used for mobile robotics, a balance needs to be made between increasing the static magnetic field, $\bar{H}$, induced in the plate, and increasing $\bar{H}$ beyond the point that it impedes the robot’s movement [18]; increasing $\bar{H}$ will increase $\bar{B}$. Optimisation options for the EMAT are limited with PPM EMATs; the constraints that have to be considered are as follows: magnet spacing determined by choice of wave mode/wavelength; magnet array layout determined by style of the PPM EMAT; practically available magnet sizing options, choosing broadband or narrowband generation/reception; and mobile robotics requirement of reduced magnetic strength to allow movement [18]. 

Using the magnet arrangement shown in Figure 3 as a reference example, the centre-to-centre spacing of magnets with the same polarity in the same row is determined by the wavelength [9]; in the example, this is 25 mm. A paper [37] by Rose carried out a study on the elements in comb style transducers for generating guided waves, which showed that when increasing the magnet width, w, over 50% of the length, S (Figure 3), little change in the generated wave amplitude is observed. However, acquiring Neodymium magnets of specific widths is not as practical as buying standard off the shelf sizes, which results in using magnets of w>S/2; this can have a small effect on performance but is typically cancelled as this will usually increase the static magnetic field, $\bar{H}$, due to the increased magnet size. It is not recommended to use magnets of width w=s as they will be difficult to manufacture and will result in increased wave harmonics, creating additional noise and no real amplitude gain compared to w>S/2 [37]. 

The width of the wavefront is determined by the outer edge to edge distance of the magnet; in the case of the example, Figure 3, this is 41 mm. The magnet width determines the width of the coil. The total width was used to calculate the beam spread of the EMAT using Equation (12) [38], where the width is aperture A, and λ is the wavelength.
(12)$$\theta =2\arcsin\left(0.442\frac{\lambda }{A}\right)$$

Six magnets were used in each of the two rows in the example EMAT. Reducing the number of magnets in a row will create broadband generation/reception and increasing the number of magnets in a row will create narrowband generation/reception [39,40]. Ideally, it is better to have more magnets for narrowband generation and reception if the EMAT is to be used at only specific wavelengths; however, this will increase the size of the EMAT, as well as the magnetic holding force, which then impacts the feasibility for use on a mobile robotic platform. There is existing work that has experimented with varying magnet sizes within a row to vary the wavelength of waves generated [29].

## 3. Validation of Optimisation Model

### 3.1. Magnetisation Validation via Experiment

Experimental comparison to the values output by the numerical model was carried out to validate the coil optimisation model. This was done by using the coil dimensions of an existing commercial standard bidirectional PPM racetrack coil EMAT, a Sonemat (Sonemat Ltd., UK) SHG2541-G, and using the model to define the number of turns and trace widths for the input dimensions given. The model determined that for the following $C_{length}=116\;\mathrm{mm}$, $C_{width}=$41 mm and $M_{width}=20\;\mathrm{mm}$, the optimal trace width was 0.63 mm and the number of turns was 18; this gives the highest magnetisation. It was found experimentally that 18 turns was the optimal number and that experiment aligned with the model.

The numerical model outputs a curve of magnetisation vs. number of turns for a given set of input parameters; this is a theoretical calculation. To validate the model, a number of test PCB coils, not FPCBs, were manufactured. Each PCB coil had different trace width values and number of turns (5, 10, 15, 18, 25); 18 turns was used instead of 20. The magnetisation values of the PCBs were then measured experimentally and compared to the values generated by the model. The resultant data is shown in Figure 4.

It can be seen that the experimental magnetisation values followed a very similar trend and closely resembled magnetisation values of the numerical model; however, there is a slight offset, which varies depending on the number of turns. The offset was likely due to over etching during the PCB manufacture process. Over etching is typically constant, which means that the same amount of copper is removed regardless of trace width. This over etching will have a greater effect on smaller traces than larger ones, which is reflected in the graph. The smaller trace widths see greater proportional increases in resistance, which reduces magnetisation.

### 3.2. Industrial EMAT Comparison

To further validate and to show the benefits of utilising the coil optimisation model, a replica of the Sonemat SHG2541-G was constructed and compared to the Sonemat transmitter. The replica transmitter has the same dimensional parameters as the Sonemat transmitter, i.e., $C_{length}$, $C_{width}$ and $M_{width}$, but a FPCB racetrack coil was used, instead of a hand wound coil, in the replica transmitter where the rest of the coil parameters were decided using the numerical model. For the coil dimensions taken from the Sonemat and used to design the replica coil, the optimal number of turns was 23 and the optimal trace width was 0.63 mm, with a copper weight of 2 oz/ft^2^. The magnet array setup for the replica was copied from the Sonemat transmitter and remained unchanged.

To compare the optimised replica and the Sonemat transmitter, both EMATs were placed on a 10 mm steel sample. The transmitters were positioned 30 cm (centre-to-centre) from a receiver EMAT and pulsed with the same voltage. The same pulsing voltage was used to illustrate the effects of optimising the racetrack coil as the optimised coil will produce higher amplitudes with the same voltage compared to the unoptimised EMAT. The experimental setup used is shown in Figure 5; a pitch-catch setup was used. A LabVIEW (NI Inc., Austin, TX, USA) program was used to control the wave generation parameters. The laptop used to run LabVIEW was connected to trigger a PicoScope 5000a (PicoTech, Cambridge, UK) to synchronise with a Ritec RPR-4000 Pulser/Receiver (Ritec, INC, Warwick, RI, USA), shown in Figure 5. The Ritec generates a 3-cycle sinusoidal pulse with a reception gain of 50 dB. A 0.5 mm lift-off was achieved using PTFE sheets. A 10 mm steel plate was used as the sample, which has a density of 7850 kg/m^3^ and a measured Longitudinal and Shear velocity of 5959 m/s and 3259 m/s.

Figure 6 shows the A-Scans of the signals generated by each transmitter at two frequencies, 128 kHz for SH0 and 205 kHz for SH1, λ = 25 mm. The clear advantages to using the optimised coil can be seen in terms of amplitude and SNR, where there is a 4 to 5 times power increase between the two EMATs in favour of the optimised replica. The replica transmitter had a 6 dB SNR increase over the Sonemat transmitter.

## 4. Mobile Robotic Transducer Design

### 4.1. Transmitter/Receiver Setup

The overarching goal of the work presented is to create an automated mobile robotic mapping system that uses guided waves as the sensory input. A two-receiver and one bidirectional transmitter setup was used [5]. This setup impedes robotic movement due to the size, approximately 30 cm in width, but is necessary due to the bidirectional transmitter, which requires two receivers so that relative feature position can be understood during ranging.

One of the benefits of using a unidirectional EMAT for ranging is that the relative position of a feature can be easily inferred. As waves are only being produced in one direction from the EMAT, there is no confusion around the wave propagation path arising from whether a wave initially travelled from the left or the right side of the transmitter. 

The result of using a unidirectional transmitter is that the overall sensor package size can be reduced significantly as there is no need for a second receiver to help determine wave propagation paths. Further to this, it was found in testing that the receiver can be placed adjacent to the transmitter and performance remains the same, as electrical saturation subsides before returning signals arrive. As this setup is pulse-echo instead of pitch-catch, electrical saturation will only affect the first received direct signal from transmitter to receiver, not those reflected from edges. Both factors allow the transmitter and receiver to be mounted in the same housing, which greatly reduces the transducer package size, which is highly advantageous for use in mobile robotics. The unidirectional transmitter/receiver setup is shown in Figure 7. 

### 4.2. Transmitter EMAT Design

A unidirectional PPM EMAT is essentially two standard PPM EMATs overlapped and integrated into each other. The magnet array of the top EMAT is side shifted a quarter wavelength ahead of the bottom EMAT [21]. Each EMAT has a coil, which is pulsed separately from the other; the top EMAT coil, $I_{2},$ is pulsed 90° after the bottom coil, $I_{1}$, which causes the two forward propagating waves to constructively interfere and the backward propagating waves to destructively interfere, essentially eliminating the wave propagation on one side [21]. 

The unidirectional EMAT designed in this work, Figure 7, was optimised for a 25 mm wavelength to generate SH1 within a 10 mm steel plate at 205 kHz. The 10 × 10 × 5 mm magnets were used as they allowed space to easily overlap the two racetrack coils and magnet arrays. The magnets were also a commercially available size. A 2.5 mm edge-to-edge spacing was created between the magnets, which means that w>S/2; however, as there are six rows of magnets, the noise created by the increased magnet size is eliminated by narrowband generation.

The FPCB coils input parameters, $G_{minimum}$, $P_{thickness}$, $T_{minimum}$ and $W_{Cu}$, were set by the manufacturer. To produce a wave with wavelength of 25 mm and to allow the coils to overlap to form a unidirectional EMAT, the rest of the input parameters are listed in Table 1.

For the unidirectional transmitter coils, the resultant trace width was 0.354 mm and the number of turns was 23; this was subsequently printed on flexible Kapton PCB with a copper weight of 2 oz/ft^2^. Although these coils are designed for a unidirectional PPM EMAT, the optimisation method presented still applies; there is no difference between the method of optimisation for this coil and a standard bidirectional coil, only the parameter inputs are different.

### 4.3. Reduced Size Receiver EMAT Design

The receiver coil was designed using the same methodology as the transmitter coils; however, the coil dimensions were altered to fit across two 20 × 10 mm magnet rows. Input parameters for the numerical model were identical except from the following, which are given with the used value: $L_{length}$—80 mm, $L_{width}$—42 mm and $M_{width}$—20 mm. This resulted in a trace width of 0.745 mm with 23 turns, printed onto flexible Kapton PCB with a copper weight of 2 oz/ft^2^.

Four magnets were used for the receiver as the decrease in magnetic holding force from using 12 magnets would greatly decrease the force impeding mobile robotic movement [18]. The receiver used in the bidirectional setup used 12 magnets. The transmitter EMAT is a narrowband generator, and as such, it has been assumed that narrowband reception was not critical, so a reduced amount of magnet rows was used. 

### 4.4. Simulation of Unidirectional Transmitter EMAT

#### 4.4.1. Simulation Setup

OnScale (OnScale, US-CA) was used to simulate the unidirectional transmitter EMAT design. A 3D model was created where blocks of the same dimensions as the magnets were used as force application areas for generating the SH wave in the simulated steel plate. The coil and the magnets were not directly simulated in OnScale, but instead, Lorentz forces were applied to the surface in the areas where the magnets and coil overlap. 

To generate SH1 in a 10 mm plate of mild steel, with a density 7850 kg/m^3^ [41], the Longitudinal and Phase velocities were 5959 m/s and 3259 m/s and a 3-cycle 205 kHz sinusoidal pulse was used as the driving signal for all four rows; however, the driving signal for rows 1 and 3 were offset by 90°. Mesh size was set to 0.1 mm, which is less than λ20 [42], so the mesh size is suitably accurate. Measurement points were set on both sides of the force array, 10 cm along the x-axis from the centre of the force array, in order to measure the dampened backward waves and the enhanced forward waves. 

#### 4.4.2. Simulation Results

Figure 8 shows the comparison in amplitude between the forward (constructive interference) and backward (destructive interference) waves; as desired, the backward wave is almost fully eliminated. In comparison to the forward travelling wave, the backward signal is in an amplitude range that could be considered noise with an amplitude less than 0.1 mV, which would reduce in amplitude through attenuation over time. As such, there is little chance of this wave producing a feature reflection, as the point at which this small signal is measured is almost next to the point of generation, and feature reflection propagation paths are typically longer. 

### 4.5. Experimental Test of Unidirectional EMAT

#### 4.5.1. Experimental Setup

To show that the optimised unidirectional transducer package operated as intended, the transmitter and receiver EMATs were tested experimentally. The transmitter test was carried out to check that single-sided wave generation occurred, and if so, the signal will resemble noise on one side and be amplified on the other. Due to equipment limitations, both coils of the unidirectional transmitter could not be pulsed at the same time and had to be pulsed separately. Post-processing was then used to phase shift the pulse of one coil by 90° and combine the signals to cause constructive and destructive interference. This also required the receiver to be moved to be in front and behind the receiver to capture the initial pulse of the forwards and backwards waves. The receiver EMAT was placed 10 cm, centre-to-centre on both sides, from the transmitter; this setup has increased transducer separation compared to the final transducer setup for experimental simplicity. 

For evaluating the receiver, an SNR comparison to a commercial receiver EMAT was used. The commercial EMAT and the reduced size receiver EMAT were placed 30 cm from the commercial transmitter EMAT. SH waves were sent to the receivers at the same voltage; the SNR of each receiver EMAT was then able to be compared. The experimental equipment setup for both tests was the same as that used in Section 4.1. As shown in Figure 5, a pitch-catch configuration was used for measurement in both cases. For the unidirectional transmitter test, the commercial EMAT, the Sonemat SHD2541-S, was used as the receiver. A Sonemat SHG2541-G was used as the transmitter for the reduced size receiver test. Both commercial EMATs are designed for a 25 mm wavelength and with a 6 × 2 magnet array made up of 20 × 10 × 5 mm magnets.

#### 4.5.2. Transmitter Test Results 

The results of the unidirectional transmitter can be seen in Figure 9, where the amplitude of the forward propagating wave was significantly increased, and the backward propagating wave was dampened greatly. 

The backward propagating wave was not fully eliminated; this is thought to be a result of the method of pulsing each coil individually and combining the signals in post-processing. However, it has still been shown that unidirectional generation was achieved using this EMAT configuration. 

#### 4.5.3. Receiver Test Results

The SNR of the commercial EMAT receiver was 23.7 dB and the SNR of the reduced size receiver was 19.4 dB. The SNR of the reduced size receiver was expected to be less than the commercial EMAT as the commercial EMAT has four more rows of magnets, which increases its reception capabilities. However, the slight reduction in SNR does not significantly affect the ability to range with the reduced size receiver, as in testing the reflections of edges and welds could still be seen even with the increase in noise. The reduced magnetic pull force is beneficial for the mobile robotic scenario this transducer setup will be used for, so the compromise of reduced SNR is acceptable. 

### 4.6. Transducer Ranging

The use of two receivers can solve the issue of determining the wave propagation path when using a bidirectional transmitter. However, the performance of this setup was not perfect and certain measurement positions within a sample would cause signal interpretation issues. An example of this would be where two features were at the same distance from the transducer setup but in opposite directions; reflecting signals would arrive at the same time in both receivers, which meant that the path the signal had travelled could not be interpreted. The unidirectional transmitter setup solves this issue as the returning waves are driven from one side of the EMAT, so the initial direction and propagation path is implied for all signals. 

Accuracy of ranging is not measured here, as in both setups, the SH1 wave mode is used, so a comparison cannot be made as this is wave mode dependant. Instead, performance will be measured by evaluating the difficulty in identifying the direction that the signal came from, and the coverage given by the number of measurements that can be taken by a transducer setup when moving in 10 cm steps across a test sample. Of the two transducer setups, the better performing setup for ranging and mapping in an unknown environment will be that which has no dead zones of identifying the received signal direction and provides increased measurement coverage of the sample.

#### 4.6.1. Mapping Performance Setup

To compare the two transducer setups, a mapping accuracy test was carried out on a 100 × 100 × 1 cm aluminium plate. An aluminium plate was used instead of steel to ease the measurement process as there are no magnetic holding forces, which eased positioning. The bidirectional setup was reduced in size so that both receivers were closer together resulting in a total transducer length of 35 cm. The unidirectional setup had a length of 15 cm, where the unidirectional transmitter and the reduced size receiver were used. 

Both setups were tested by mounting them to a robotic crawler and driving the crawler laterally across the sample from one side to the other, moving the transducers and recording the ease of identifying the reflected signal and the certainty of the direction of the signal (Figure 10). A housing was created for both setups to mount to the robotic crawler and to maintain the same relative transmitter and receiver position(s) during measurement. Starting at the edge of the plate, measurements were taken as many times as possible in 10 cm steps; however, due to the different sizes of transducer, a different number of measurements were taken for each. 

#### 4.6.2. Mapping Performance Results

Comparing the total number of measurements taken with each setup, extra coverage was achieved within the 1 m plate using the unidirectional configuration. This is beneficial as it increases the amount of subject data available during inspection. There were two positions along the plate where it was difficult to identify an edge reflection using the unidirectional transducer setup. This was the penultimate and final measurement position, where the wave travelled from one end of the plate and back. As a result of the wave travelling this distance, attenuation had a significant effect on the returning signal. However, in a mapping situation, the transducer position is changing constantly, and this issue would be resolved. 

For the two receivers and bidirectional transmitter setup, issues were present in this experiment. There were positions where two reflecting signals arrived at the same time to each receiver, and therefore, could not be distinguished, resulting in the direction of propagation not being established. It can be seen in the signals, shown in Figure 9, that the signal direction of the unidirectional EMAT setup is known and the distance to the edge is calculated. For the signals received by the bidirectional transducer setup, as shown in Figure 11, the direction that the signals came from cannot be interpreted, and therefore, ranging cannot occur in an unknown environment. 

Further to this, there were a number of positions where identification of the reflecting signal and direction of its propagation were difficult to interpret, these additional scans are shown in Figure 12.

As a result of this successful ranging experiment, the unidirectional transmitter setup is proposed for use in future work for the guided wave mobile robotic mapping.

## 5. Conclusions

To further EMAT enabled range finding for mobile robot localisation, the bidirectional transmitter, two-receiver setup [5] used in the authors previous study was improved upon in this study by replacing it with an optimized unidirectional PPM transmitter setup.

The first improvement was made by increasing the generated Lorentz force by the EMAT to improve the quality of the signal produced at lower voltages; this was achieved by building a numerical model for racetrack coil parameter optimisation based on desired input parameters and by creating a guideline for magnet array construction and design, which was experimentally validated and compared to an existing commercial transmitter. A replica EMAT of a commercial transmitter built using the optimization methodology produced waves with a 4–5 fold power increase over the commercial transmitter when pulsed with the same voltages. 

A unidirectional transmitter-based transducer setup was proposed as an improvement over the existing bidirectional setup as it reduces the overall size of the transducer setup by removing the need for a second receiver as the initial direction of the transmitted signal is always known.

This unidirectional transmitter was manufactured using the EMAT optimisation framework along with a reduced size receiver to complete the transducer setup, resulting in a compact package better suited to small robotic crawlers with increased signal amplitude. This was tested experimentally for proof of concept. 

Finally, the two setups were compared by their ranging capabilities. The two setups were moved across an aluminium plate and the number of 10 cm measurement steps where the direction of the reflected signals could be implied and the reflected edges were easily detectable were counted. The unidirectional setup was able to range effectively at almost all points, becoming ineffective at one point only due to wave mode attenuation, as well as being able to take more measurements due to its reduced size. The bidirectional transmitter setup was unable to range at certain points due to the signals returning to the receivers at the same time, so signal direction could not be inferred. 

Future work will include further improving the dampening of the backward propagating signal so that there are no small residual signals produced. This transducer system will then be implemented onto the robotic crawler system under development, so that it can be used for mapping applications, as its effectiveness has been shown. Additionally, further investigation into coil parameter optimisation and magnet shape will be carried out.

## Figures and Tables

**Figure 1 sensors-23-02012-f001:**
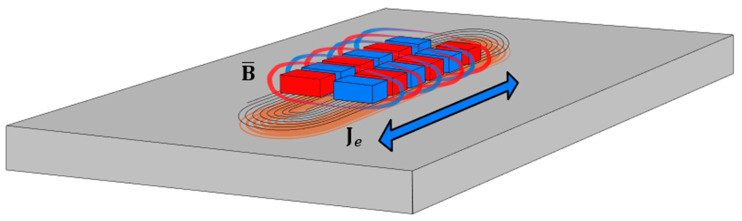
Lorentz force diagram of a typical bidirectional PPM EMAT. Shown are the induced eddy currents in orange, magnetic flux density surrounding the magnets and the direction of propagation of the generated guided waves depicted by the blue arrow.

**Figure 2 sensors-23-02012-f002:**
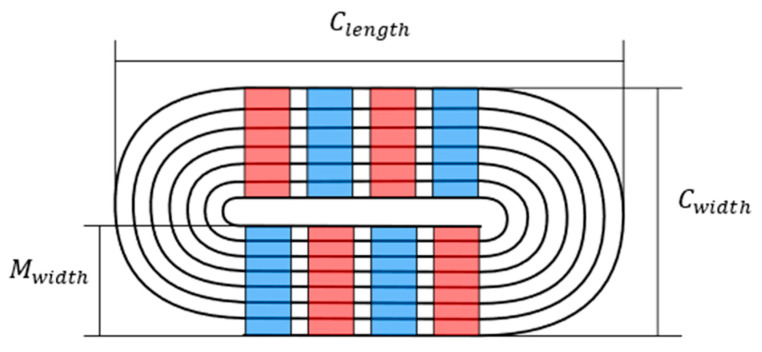
Geometrical coil input parameters for the numerical model, shown diagrammatically.

**Figure 3 sensors-23-02012-f003:**
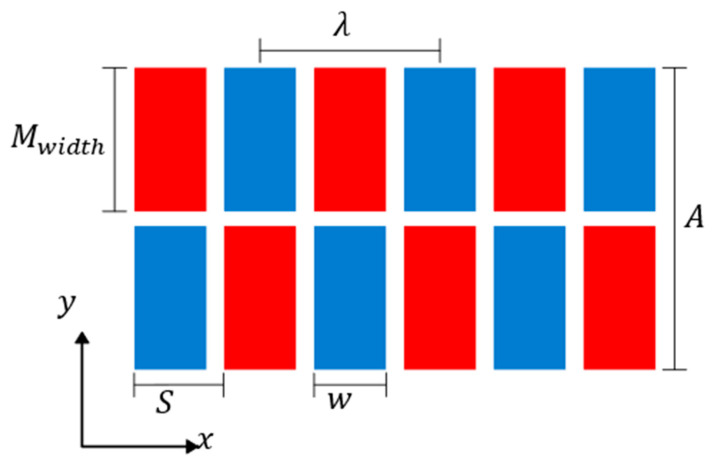
Diagram showing the magnet array spacing dimensions for PPM EMAT based on [37,38]. The red and blue rectangles represent the magnets and their polarization.

**Figure 4 sensors-23-02012-f004:**
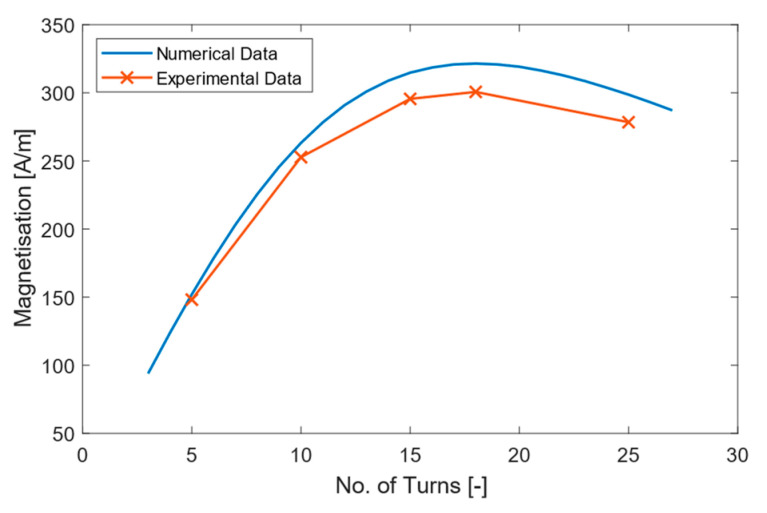
Numerical model magnetisation. Number of turns curve, blue, with the experimental data points for 5, 10, 15, 18 and 25 turns shown, orange.

**Figure 5 sensors-23-02012-f005:**
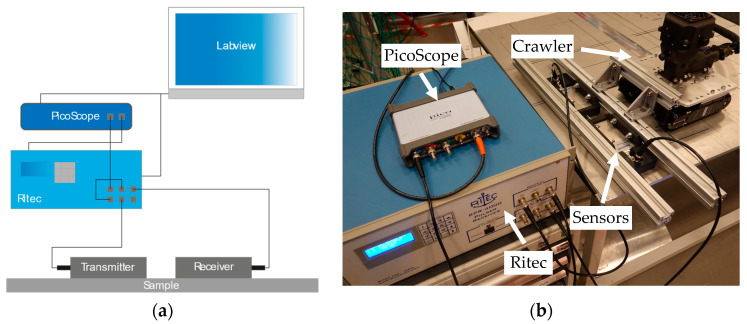
Experimental setup, shown diagrammatically (**a**) and practically (**b**). The Ritec pulses the EMATs and receives the resultant signals. The Picoscope is triggered to record this and feeds the information to the LabVIEW program so it can be analysed.

**Figure 6 sensors-23-02012-f006:**
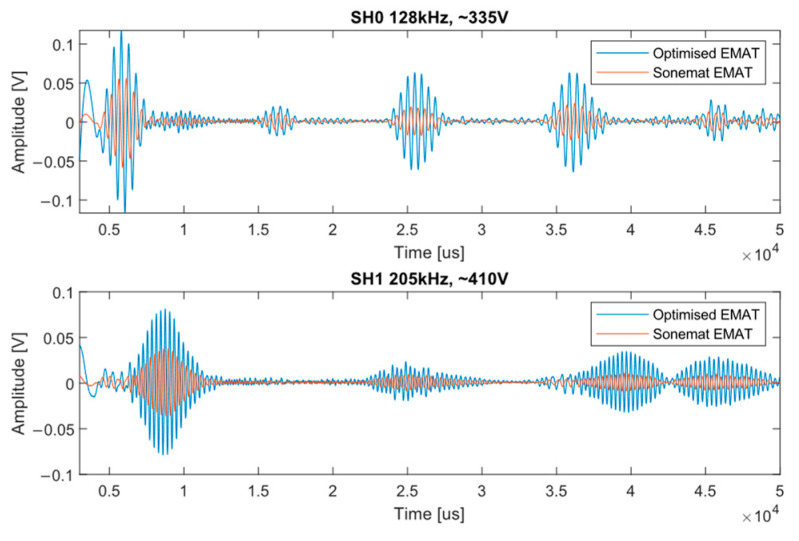
The top graph shows the A-Scans of each transmitter when generating SH0 and the bottom graph shows the A-Scans for SH1. It can be seen clearly the increase in signal amplitude when using the optimised transmitter.

**Figure 7 sensors-23-02012-f007:**
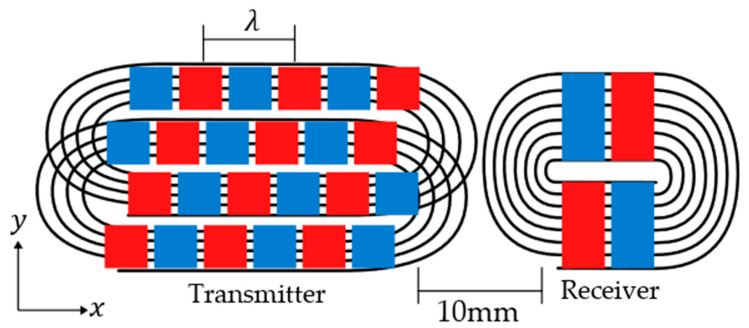
Diagram showing the magnet and coil arrangement of the unidirectional reduced receiver setup. 10 × 10 mm magnets were used for the transmitter and 20 × 10 mm magnets were used for the receiver.

**Figure 8 sensors-23-02012-f008:**
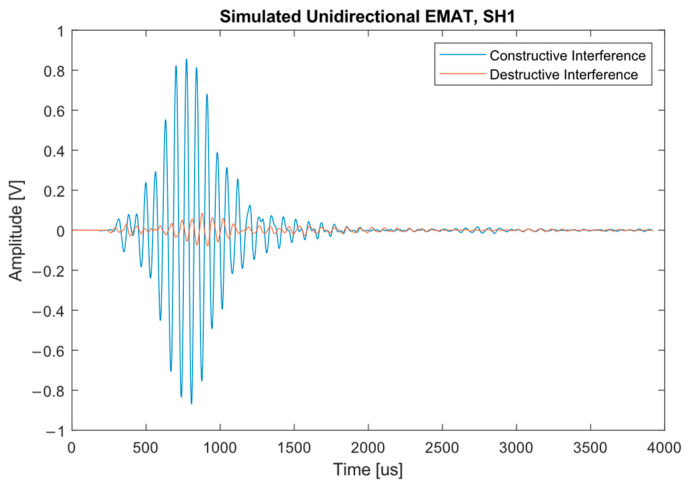
Signals produced by the simulated unidirectional transmitter. The dampened wave is significantly reduced in size compared to the undampened wave.

**Figure 9 sensors-23-02012-f009:**
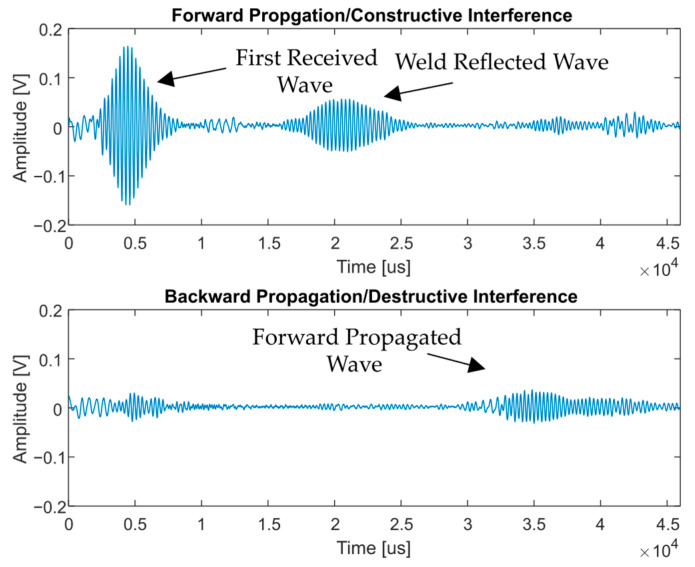
The forward propagating waves and the backward propagating waves are shown. The difference in amplitude shows that unidirectional transmission has been achieved experimentally. The signal at 3.5 µs in the backward propagating signal is a reflected signal from the forward propagating wave.

**Figure 10 sensors-23-02012-f010:**
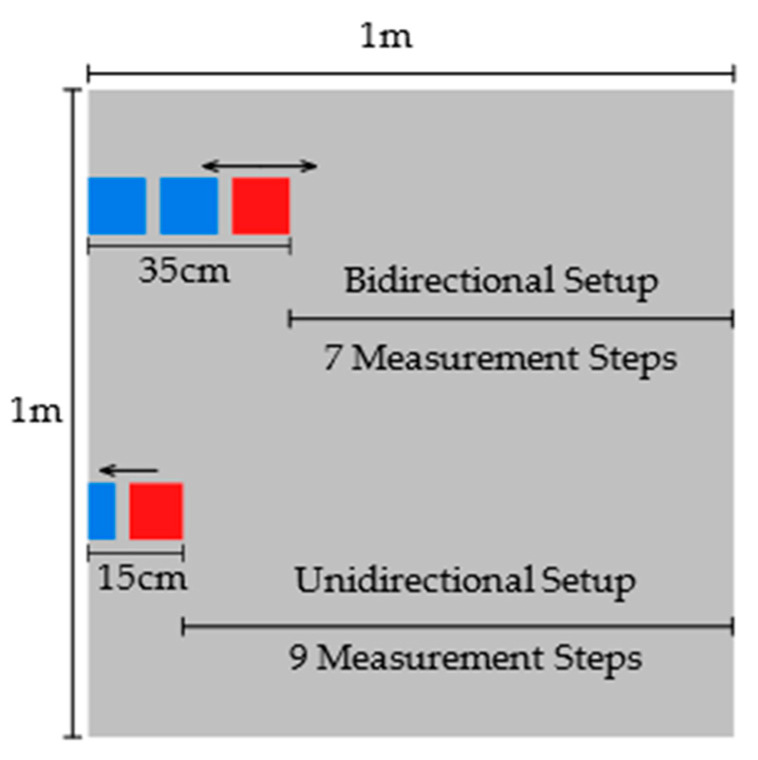
Ranging test setup and plate, the number of 10 cm steps for each transducer setup is illustrated, the bidirectional transducer setup is 35 cm in length and unidirectional transducer setup is 15 cm. Transmitters are represented by the red squares and receivers by the blue squares. The arrows indicate the directions of wave generation.

**Figure 11 sensors-23-02012-f011:**
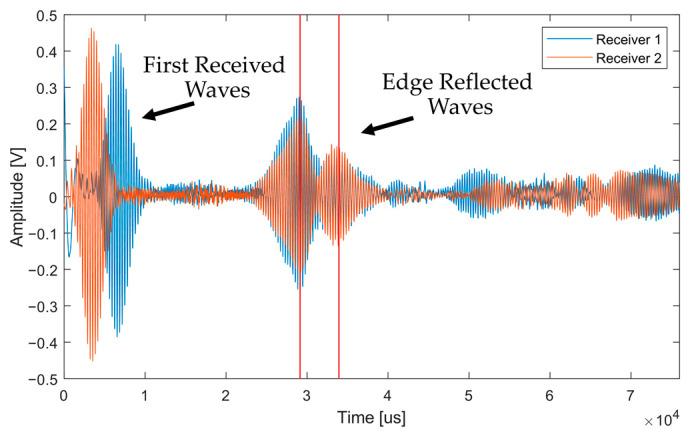
An example of a point on the plate where the direction of the signals could not be inferred due to the reflecting signals returning back to both receivers at the same time, as indicated by the red lines.

**Figure 12 sensors-23-02012-f012:**
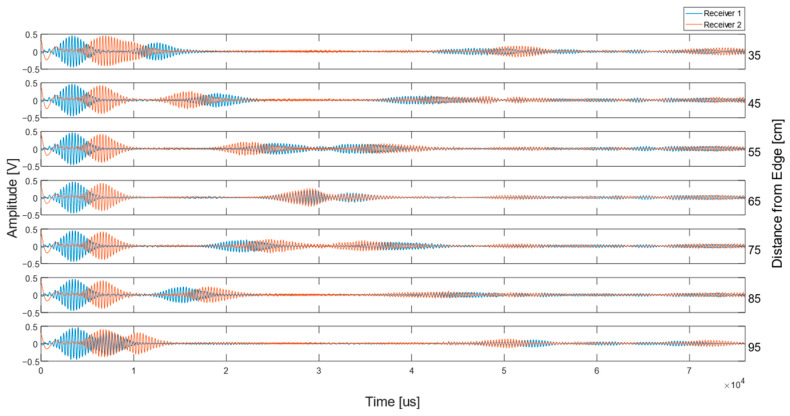
Shows the entire seven scans taken to further highlight the point on the plate where the two-receiver setup cannot map; this is 65 cm from the plate edge.

**Table 1 sensors-23-02012-t001:** Parameter input values for the unidirectional transmitter coils.

Parameter	Value	Parameter	Value
$C_{length}$	116 mm	$P_{thickness}$	0.25 mm
$C_{width}$	36 mm	$G_{sample}$	0.5 mm
$M_{width}$	10 mm	$V_{in}$	1600 V
$G_{minimum}$	0.25 mm	$Z_{source}$	50 Ω
$T_{minimum}$	0.25 mm	f	205 kHz
$W_{Cu}$	2 oz/ft^2^	$S_{M}$	Steel

## Data Availability

Data available on request, contact corresponding author.

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
