# Peer review of "Design and Manufacture of an Optimised Side-Shifted PPM EMAT Array for Use in Mobile Robotic Localisation"

_sensors, 2023, doi:10.3390/s23042012_

Round 1
Reviewer 1 Report
A potentially useful methodology for optimising EMAT coil designs which seems to work in this case. A few minor comments:
pg2: did you define SNR?
pg3: The magnetic flux lines are faint and hard to follow on the figure. Also the grammar seems worse on this page for some reason.
pg6: (just an observation): - no allowance is made for the effect of standoff on the impedence of the coil. The effect is not neglegible.
8 - are these the correct part number for Sonemat EMATs? the third letter designates generator (G) or detector (D). The last letter designates the encapsulation, which I think in your case was "S" for stainless on both EMATs.
9- I wonder how performance varies with standoff with both designs (I appreciate it is probably beyond the scope of this paper)
14: Line 543. Transducer Pose. Do you mean position?
15: It would have been nice to see the full set of results from the scans, e.g. in a "B-scan" format. It would give context to the results you did show.
Author Response
"Please see the attachment."

Reviewer 2 Report
I would like to thank the editor to give me the opportunity to review this interesting work. The impression of the paper is interesting. I have a minor comment about the future studies, which will guide the researchers to continue this good work.
The article is well written and easy to understand. However, few of my feedback can be considered to improve the quality of the paper.
1. Introduction may be improved, adding the highlights and the problem statements.
2. You could improve writing, link better the ideas flow in the Introduction.
3. Review references because some of them are unstandardized.
4. The conclusion needs improvements towards major claimed contribution.
5. Write some future directions in the conclusion section.
6. The difference between your proposal and related works is not clear, you could to details better. I suggest add a comparative table in ''Related Literature'' to contrast your solution in front of related works.
7. You could discuss the relationship between your solution and past literature.
8. In addition to the typical design, the author clearly expresses the coil selection and calculation solution. However, the description of how to position the mobile robot is not clear. Can the design of the positioning and the results of the experiment be clearly stated?
9. Referring to the experimental design in Figure 5, can you explain the connection with a section of robot? And the physical image of actual operation?
10. The conclusion should not be put on the graph, the graph should be put on the front for discussion.
11. The control principle of the crawling of small robots has not been described, please explain.
